# Leveraging MLLM Embeddings and Attribute Smoothing for Compositional Zero-Shot Learning

## Abstract

Compositional zero-shot learning (CZSL) aims to recognize novel compositions of attributes and objects learned from seen compositions. Previous works disentangle attribute and object by extracting shared and exclusive parts between image pairs sharing the same attribute (object), as well as aligning them with pretrained word embeddings to improve unseen attribute-object recognition. Despite the significant achievements of existing efforts, they are hampered by three limitations: (1) the efficacy of disentanglement is compromised due to the influence of the background and the intricate entanglement of attribute with object in the same parts. (2) existing word embeddings fail to capture complex multimodal semantic information. (3) overconfidence exhibited by existing models in seen compositions hinders their generalization to novel compositions. Being aware of these, we propose a novel framework named Multimodal Large Language Model (MLLM) embeddings and at**TR**ibute smooth**I**ng gui**DE**d dise**NT**anglement (**TRIDENT**) for CZSL. First, we leverage feature adaptive aggregation (FAA) modules to mitigate the impact of background, and utilize learnable condition masks to capture multi-granularity features for subsequent disentanglement. Then, the last hidden states of MLLM are employed as word embeddings for their superior representation capabilities. Moreover, we propose attribute smoothing through leveraging auxiliary attributes generated by Large Language Model (LLM) for each seen composition, addressing the issue of overconfidence by encouraging the model to learn more attributes in one given composition instead of just fitting a fixed attribute-object combination. Extensive experiments demonstrate that **TRIDENT** achieves state-of-the-art performance on three challenging datasets: MIT-States, C-GQA, and VAW-CZSL, respectively.

## 1 Introduction

As for the study of compositional generalization ability inherent to humans, compositional zero-shot learning (CZSL) (Misra et al., 2017; Nagarajan & Grauman, 2018; Purushwalkam et al., 2019) is proposed to enable machines to recognize unseen attribute-object compositions by leveraging knowledge of attributes and objects (*i.e.*, primitives) learned from seen compositions. Specifically, in the training phase, models are provided with images and compositional labels (*e.g.*, `ripe orange` and `peeled apple`). During the testing phase, given an image depicting a novel composition (*e.g.*, `peeled orange`), models are assigned to classify the image into the corresponding category (Zhang et al., 2022b).

Prior works (Misra et al., 2017; Nan et al., 2019) focus on mapping the visual features and the word embeddings of compositions into a joint space. These methods have poor generalization capability to unseen compositions, as they fail to learn primitives. Therefore, recent studies (Saini et al., 2022; Hao et al., 2023; Li et al., 2024) consider visual disentanglement. Among them, some prominent works deploy a triplet of images to disentangle: a given image (noted as the main image), and two supplementary images, each sharing either the same attribute or the same object as the main image. The triplet of images is treated as two image pairs for subsequent analysis. These approaches aim to disentangle attribute and object by analyzing the shared and exclusive features of the image pair, as well as aligning them with word embeddings (*e.g.*, GloVe (Pennington et al., 2014)), as shown in Figure 1. Although these pioneer research studies have achieved great progress, they exhibit three limitations:

Figure 1: A general comparison between the existing method and our proposed **TRIDENT**. Note that, we only present the representation learning of an image pair sharing the object for brevity.

**L1:** Disentanglement is impeded due to the influence of the background and the intricate entanglement of attribute with object in the same parts of image. On the one hand, models tend to extract the background feature unique to one image in the pair as the disentangled exclusive features. On the other hand, some existing methods (Ruis et al., 2021; Saini et al., 2022) compute the similarity of image pairs for disentanglement at the spatial level. However, disentangling attribute and object at the spatial level presents significant challenges because they entangle in the same spatial features. Taking an image of `ripe apple` as an example, the spatial regions corresponding to the attribute "`ripe`" and the object "`apple`" are fully co-located.

**L2:** Existing word embeddings lack the depth needed to capture complex multimodal semantic information. To begin with, word embeddings, such as Word2Vec (Mikolov, 2013) and GloVe (Pennington et al., 2014), are grounded in word frequency and contextual co-occurrence, rather than capturing high-level semantic nuances(Sarzynska-Wawer et al., 2021). Moreover, the process of aligning visual features with word embeddings can be viewed as a form of cross-modal matching; however, these word embeddings are trained only in a single text modal, preventing them from capturing cross-modal information between images and texts.

**L3:** Existing methods display excessive confidence in seen compositions, impairing their ability to generalize toward novel compositions. Due to the one-hot label used during training, these approaches are limited by learning only one attribute and object, neglecting the fact that objects naturally exhibit multiple attributes (Xu et al., 2024). Consequently, models exhibit overconfidence in the disentangled ground-truth attribute, treating other attributes that can describe the object as negative attributes, which results in the diminished performance on unseen compositions.

To address the aforementioned limitations, we propose a novel framework named Multimodal Large Language Model (MLLM) embeddings and at**TR**ibute smooth**I**ng gui**DE**d dise**NT**anglement (**TRIDENT**), which consists of three major modules: visual feature extraction, attribute-object disentanglement, and feature alignment. The first module leverages feature adaptive aggregation (FAA) modules to mitigate the impact of background noise, and exploits learnable condition masks to learn multi-granularity features to improve subsequent disentanglement. The second module aims at leveraging shared and exclusive weights of image pairs to disentangle attribute and object under the the paradigm that apart from the shared features of the image pair, each image has its own exclusive features. The third module is intended to align the visual features of compositions and disentangled primitives with the last hidden states of an MLLM, LLaVA v1.5 (Liu et al., 2024a), *i.e.*, MLLM embeddings. This is inspired by some works (Wang & Kuo, 2020; Muennighoff, 2022; Muennighoff et al., 2024; Koh et al., 2023), which find that the last hidden states of (M)LLM exhibit powerful representational capabilities in embedding tasks, such as retrieval and classification. Moreover, to tackle the issue that the ineffective overconfidence of the models regarding ground-truth attribute hinders them from generalizing to unseen compositions, we exploit a Large Language Model (LLM), GPT-3.5 (OpenAI, 2023) to generate auxiliary attributes based on attribute-object compositions and perform label smoothing for attributes, *i.e.*, attribute smoothing.

In summary, the main contributions of our work are three-fold:

1. We propose novel feature adaptive aggregation modules to reduce the impact of background, and utilize learnable condition masks to capture multi-granularity features for disentanglement in CZSL.

2. We employ both LLM and MLLM to guide attribute-object disentanglement by generating auxiliary attributes and representing word embeddings, respectively. To the best of our knowledge, we are the first to leverage both LLM and MLLM to advance disentanglement in CZSL task.

3. We conduct extensive experiments to evaluate our method on three CZSL benchmarks, showing that **TRIDENT** has achieved state-of-the-art performance.

## 2 RELATED WORK

**Compositional zero-shot learning (CZSL).** Prior works in CZSL can be broadly divided into two main streams. One main stream is to learn representations of compositions in a joint space. SymNet (Li et al., 2020) proposes to learn symmetry property in compositions. Co-CGE (Mancini et al., 2022) leverages a Graph Convolutional Neural Network to learn compositional representations. The other main stream aims at disentangling visual representations of primitives to reduce composition learning into primitive learning. SCEN (Li et al., 2022) leverages contrastive loss to excavate discriminative prototypes of primitives. INV (Zhang et al., 2022b) learns domain-invariant primitives. OADis (Saini et al., 2022) and ADE (Hao et al., 2023) disentangle primitives by affinity modules and the multi-head attention mechanism, respectively. CANet (Wang et al., 2023) learns conditional attribute conditioned on the recognized object and the input image.

More recent works (Nayak et al., 2023; Lu et al., 2023; Huang et al., 2024) focus on leveraging the encyclopedic knowledge of pretrained vision-language models (VLM), such as Contrastive Language-Image Pre-training (CLIP) (Radford et al., 2021) and Context Optimization (CoOp) (Zhou et al., 2022), to encode and align images and texts.

**Large language model (LLM).** LLMs have realized significant advancements thanks to the scaling up of training data and the increase in the number of parameters. Early models, such as BERT (Devlin et al., 2019) and GPT-2 (Radford et al., 2019), initially exhibit strong capabilities in understanding and generating human-like language. Subsequently, GPT-3 (Brown et al., 2020) with about 175 billion parameters demonstrates great breakthroughs across numerous language benchmarks. This development has facilitated the emergence of many LLMs, including OPT (Zhang et al., 2022a) and LLaMA (Touvron et al., 2023). Moreover, by performing instruction fine-tuning on LLM, ChatGPT (Ouyang et al., 2022) and Vicuna (Chiang et al., 2023; Zheng et al., 2023) are competent at comprehending and following human instructions better.

Expanding on LLM, Multimodal Large Language Model (MLLM) incorporates a pretrained visual encoder for vision-language tasks. Flamingo (Alayrac et al., 2022) first integrates Vision Transformer (ViT) (Dosovitskiy, 2020) and LLM by gated cross-attention. BLIP-2 (Li et al., 2023) puts forward a Q-Former module to bridge the modality gap. Recently, LLaVA (Liu et al., 2024b) and LLaVA v1.5 (Liu et al., 2024a) introduce visual instruction tuning to enhance instruction following capability. The visual understanding part of LLaVA v1.5 consists of a ViT and a multilayer perceptron (MLP) cross-modal connector (CMC). CMC processes visual features before the last layer of ViT, aligning the visual space of ViT with the linguistic space of LLM. We choose LLaVA v1.5 as our foundational MLLM as it has demonstrated state-of-the-art performance across various tasks.

Recently, exploring the powerful language capabilities of (M)LLM to handle representation tasks (*e.g.*, retrieval) has emerged as a prominent research domain. SGPT (Muennighoff, 2022) exploits the last hidden states of LLM for the input token sequence or a special learnable token to derive representational embeddings. Subsequently, GritLM (Muennighoff et al., 2024) applies mean pooling over the last hidden states of LLM to yield the textual embeddings. FROMAGe (Koh et al., 2023) leverages a learnable [RET] token to represent the content fed into MLLM for image retrieval.

## 3 APPROACH

### 3.1 TASK FORMULATION

Compositional zero-shot learning (CZSL) aims at learning a model that can recognize unseen compositions of attributes and objects that are learned from seen compositions. Given an attribute set $\mathbb{A}$ and an object set $\mathbb{O}$, the attributes and objects are composed to form a composition set $\mathbb{C} = \mathbb{A} \times \mathbb{O}$. The composition set $\mathbb{C}$ is divided into two disjoint sets: the seen composition set $\mathbb{C}_s$ and the unseen composition set $\mathbb{C}_u$, where $\mathbb{C}_s \cap \mathbb{C}_u = \varnothing$ and $\mathbb{C}_s \cup \mathbb{C}_u = \mathbb{C}$. The model is trained with a seen training set $\mathbb{D}_{tr} = \{(x_s, c_s)\}$, where $x_s \in \mathbb{X}_s$ is an image from the seen image set $\mathbb{X}_s$ corresponding to the seen composition set $\mathbb{C}_s$, and $c_s \in \mathbb{C}_s$ is the label of $x_s$. Following the Generalized CZSL (Purushwalkam et al., 2019), the model is evaluated on a predefined test set $\mathbb{D}_{te} = \{(x_{te}, c_{te})\}$, where $x_{te} \in \mathbb{X}_{te}$ is an image from the unseen image set $\mathbb{X}_{te}$ corresponding to the composition subset $\mathbb{C}_{te}$ of $\mathbb{C}$, *i.e.*, $\mathbb{C}_{te} \subseteq \mathbb{C}$, and $c_{te} \in \mathbb{C}_{te}$ is the label of $x_{te}$. The aim of CZSL task is to learn a model $M : \mathbb{X}_{te} \rightarrow \mathbb{C}_{te}$ that predicts labels $c_{te}$ from $\mathbb{C}_{te}$ for the input images $x_{te} \in \mathbb{X}_{te}$.

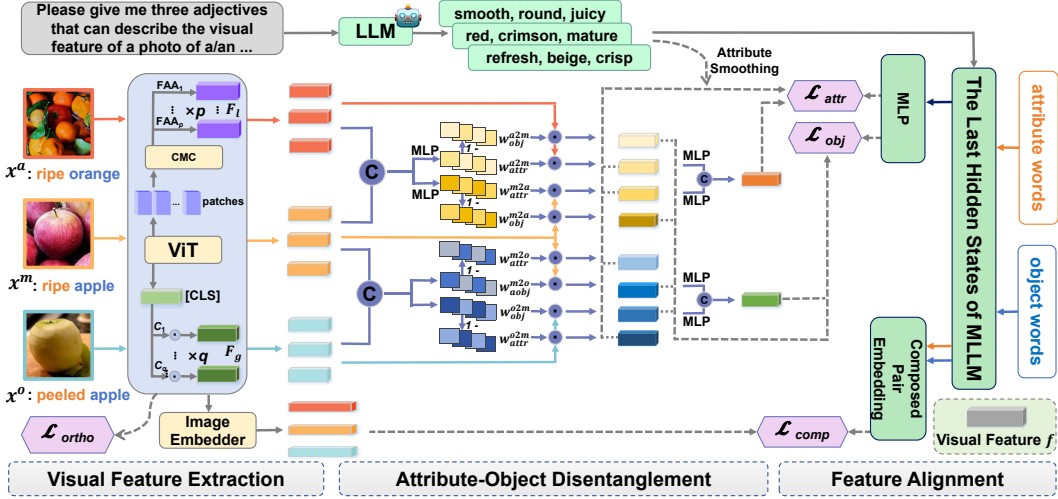

Figure 2: The overall architecture of our proposed **TRIDENT**. **TRIDENT** consists of three major modules: (a) visual feature extraction, (b) attribute-object disentanglement, and (c) feature alignment.

## 3.2 TRIDENT

As the major novelty, we propose a novel framework named MLLM embeddings and attribute smoothing guided disentanglement framework (**TRIDENT**) for CZSL, as shown in Figure 2. It consists of three major modules: (1) visual feature extraction, (2) attribute-object disentanglement, and (3) feature alignment. We now detail each module of **TRIDENT** in this section.

### 3.2.1 VISUAL FEATURE EXTRACTION

As shown in Figure 2, we denote a given image with the attribute-object composition label (e.g. `ripe apple`) as the main image $x^m$, and randomly sample an image with the same attribute $x^a$ (*i.e.*, `ripe orange`), as well as an image sharing the same object $x^o$ (*i.e.*, `peeled apple`) to comprise a triplet image set. For the convenience of expression, we simply use $x^{img}$ (where $img \in \{m, a, o\}$) to collectively denote the images as they are processed using the same module.

**Visual feature extraction backbone.** As mentioned before, since LLaVA v1.5 is used as our fundamental MLLM, we directly leverage the visual encoder, ViT, and cross-modal connector (CMC) from the model to extract visual features. Specifically, the image $x^{img}$ is partitioned into $n$ patch tokens, which are subsequently put into ViT along with the `[CLS]` token. Afterward, the output of patch tokens before the last layer of ViT is fed into the CMC module, as implemented in LLaVA v1.5. To align the dimension of patch tokens output by CMC with that of `[CLS]` token produced by ViT, the patch tokens output by CMC are input into a linear layer. Consequently, we obtain one feature vector of `[CLS]` token $f_{cls}^{img} \in \mathbb{R}^d$ and a patch feature matrix of $n$ patch tokens $F_{patch}^{img} \in \mathbb{R}^{n \times d}$, where $d$ is the dimension of the features.

**Local features extraction.** Intuitively, the composition (*e.g.*, `ripe apple`) only occupies a few parts of the image. Since each patch token usually corresponds to one local region of the image, to filter out background noise and focus on related regions, we deploy a set of feature adaptive aggregation (FAA) modules to derive $p$ relevant local features of $x^{img}$, where each FAA module is formulated as follows:

$$v = a \otimes F_{patch}^{img} \quad with \quad a = \sigma(Conv(F_{patch}^{img})) \tag{1}$$

where $Conv(\cdot)$ represents the $1 \times 1$ convolution layer, $\sigma(\cdot)$ denotes the sigmoid activation function, $a \in \mathbb{R}^n$ is the weight vector, the $k$-th element of $a$ is the weight for $k$-th patch feature. $\otimes$ represents matrix product, and $v \in \mathbb{R}^d$ is the local feature obtained by an FAA module. Subsequently, we vertically concatenate the local features produced by $p$ FAA modules to obtain the local feature matrix $F_l^{img} \in \mathbb{R}^{p \times d}$.

**Global features extraction.** Normally, the ViT output of [CLS] token is regarded as containing various global information of the image, which highly entangles both attribute and object features together(Hao et al., 2023). To disperse multi-granularity global information into different representations, $q$ learnable condition masks are applied to $\boldsymbol{f}_{cls}^{img}$ to obtain $q$ different global representations, where each global representation is computed as:

$$\boldsymbol{u} = \boldsymbol{f}_{cls}^{img} \odot \boldsymbol{c} \tag{2}$$

where $\boldsymbol{u} \in \mathbb{R}^d$ denotes each global representation. Here $\boldsymbol{c} \in \mathbb{R}^d$ refers to each learnable condition mask and $\odot$ is the element-wise multiplication. Consequently, we vertically concatenate $q$ global representations to derive the global feature matrix $\boldsymbol{F}_g^{img} \in \mathbb{R}^{q \times d}$.

**Features concatenation.** Finally, both $\boldsymbol{F}_l^{img}$ and $\boldsymbol{F}_g^{img}$ are vertically concatenated to form the visual features of $x^{img}$, i.e., $\boldsymbol{F}^{img} = [\boldsymbol{F}_l^{img}, \boldsymbol{F}_g^{img}] \in \mathbb{R}^{(p+q) \times d}$, which is used for the following disentanglement of attribute and object.

**Orthogonal regularization.** We ideally want features extracted by different modules can represent different information of the image $x^{img}$. To this end, we further introduce the orthogonal regularization, i.e.:

$$\mathcal{L}_{ortho} = \sum_{img \in \{m,a,o\}} (\|\boldsymbol{F}^{img} \boldsymbol{F}^{img^T} - \boldsymbol{I}\|_{Fro}) \tag{3}$$

where $\boldsymbol{I} \in \mathbb{R}^{(p+q) \times (p+q)}$ is the identity matrix, and $\| \cdot \|_{Fro}$ refers to the Frobenius norm of the matrix.

**Image embedder.** Inspired by Nagarajan & Grauman (2018), for the input image $x^{img}$, we first use AveragePools $Avg(\cdot)$ on $\boldsymbol{F}_g^{img}$ and $\boldsymbol{F}_l^{img}$, respectively, and horizontally concatenate them by $Cat(\cdot, \cdot)$ to aggregate both global and local visual information of $x^{img}$ corresponding to the composition label. Then the concatenated feature passes through a linear layer $Lin_{comp}(\cdot)$ to derive the final feature representation $f_{comp}^{img}$. This module is formulated as follows:

$$\boldsymbol{f}_{comp}^{img} = Lin_{comp}(Cat(Avg(\boldsymbol{F}_g^{img}), Avg(\boldsymbol{F}_l^{img}))) \tag{4}$$

where $\boldsymbol{f}_{comp}^{img} \in \mathbb{R}^{2d}$ denotes the visual feature corresponding to the composition. This module is designed to learn the visual features of images associated with their corresponding composition labels, serving as the primary branch for inference.

### 3.2.2 ATTRIBUTE-OBJECT DISENTANGLEMENT

As mentioned before, one of the key challenges for CZSL task is to disentangle attribute and object from visual features. To overcome such challenge, we propose a novel weighted disentanglement module to disentangle primitives, as illustrated in Figure 2. For brevity, one image pair $x^m$ and $x^a$ from the triplet image set is taken as an example to elaborate on this module, while another image pair $x^m$ and $x^o$ follows the same architecture.

**Weights computation.** The features of $x^m$ and $x^a$ (i.e., $\boldsymbol{F}^a$ and $\boldsymbol{F}^o$) are vertically concatenated and fed into two MLP modules to derive their respective weights of shared attribute features relative to each other, and subsequently utilize them to compute the weights of their own exclusive object features as follows:

$$\begin{cases} \boldsymbol{w}_{attr}^{m2a} = \sigma(MLP_{m2a}([\boldsymbol{F}^m, \boldsymbol{F}^a])) \\ \boldsymbol{w}_{obj}^{m2a} = \boldsymbol{1} - \boldsymbol{w}_{attr}^{m2a} \\ \boldsymbol{w}_{attr}^{a2m} = \sigma(MLP_{a2m}([\boldsymbol{F}^m, \boldsymbol{F}^a])) \\ \boldsymbol{w}_{obj}^{a2m} = \boldsymbol{1} - \boldsymbol{w}_{attr}^{a2m} \end{cases} \tag{5}$$

where $\boldsymbol{w}_{attr}^{m2a}, \boldsymbol{w}_{attr}^{a2m} \in \mathbb{R}^h$ demonstrate the weights of the shared attribute features of $x^m$ relative to $x^a$, and $x^a$ relative to $x^m$, respectively. $\boldsymbol{w}_{obj}^{m2a}$ and $\boldsymbol{w}_{obj}^{a2m}$ denote the weights of exclusive object features corresponding to $x^m$ and $x^a$, respectively, which are derived by "$1 - shared\ weights$" paradigm as beyond the shared features of the image pair are the exclusive features of each image. Taking $\boldsymbol{w}_{attr}^{m2a}$ as an example, its $k$-th element refers to the shared attribute proportion of $k$-th feature of $x^m$ relative to $x^a$.

**Disentangled features obtainment.** We multiply elements of each weight by the corresponding features and then calculate the average. The following takes the process of obtaining the shared attribute features of image $x^m$ relative to $x^a$ as an example:

$$\boldsymbol{f}_{attr}^{m2a} = \frac{1}{h} \sum_{i=1}^{h} w_{attr\ i}^{m2a} \ \boldsymbol{F}_{i,:}^{a} \tag{6}$$

where $\boldsymbol{F}_{i,:}^{a}$ denotes the $i$-th row of $\boldsymbol{F}^a$, $i.e.$, the $i$-th feature of $x^a$. $w_{attr\ i}^{m2a}$ refers to the $i$-th element of $\boldsymbol{w}_{attr}^{m2a}$, and $\boldsymbol{f}_{attr}^{m2a} \in \mathbb{R}^d$ is the shared attribute feature of $x^m$ relative to $x^a$.

For the image pair of $x^m$ and $x^a$, four parts are obtained: the shared attribute features of $x^m$ relative to $x^a$, and $x^a$ relative to $x^m$, $i.e.$, $\boldsymbol{f}_{attr}^{m2a}$ and $\boldsymbol{f}_{attr}^{a2m}$, as well as the exclusive object features of $x^m$ and $x^a$, respectively, $i.e.$, $\boldsymbol{f}_{obj}^{m2a}$ and $\boldsymbol{f}_{obj}^{a2m}$. Then the shared attribute feature of $x^a$ and $x^m$ without relativity $\boldsymbol{f}_{attr}^{ma}$ is obtained by an MLP layer, which is less dependent on the object. The process is as follows:

$$\boldsymbol{f}_{attr}^{ma} = MLP_{ma}(Cat(\boldsymbol{f}_{attr}^{m2a}, \boldsymbol{f}_{attr}^{a2m})) \tag{7}$$

Similarly, we disentangle attribute and object for $x^m$ and $x^o$ and obtain the same processed features as $x^m$ and $x^a$: $F_{obj}^{m2o}, F_{obj}^{o2m}, F_{attr}^{m2o}, F_{attr}^{o2m}$, and $F_{obj}^{mo}$.

### 3.2.3 FEATURE ALIGNMENT

Inspired by Muennighoff et al. (2024) that leverages the last hidden states as the representation embeddings, we consider the last hidden states of LLaVA v1.5 (Liu et al., 2024a) as our MLLM embeddings for words. Moreover, to tackle the problem that the ineffective overconfidence exhibited by the models in terms of the ground-truth attribute hinders them from generalizing to unseen compositions, GPT 3.5 is employed to generate several auxiliary attributes that describe an object with only one ground-truth attribute and perform label smoothing during attribute alignment. Now we detail each part of feature alignment.

**Generating auxiliary attribute words by LLM.** Since only attribute text needs to be generated, we leverage a LLM, GPT-3.5, instead of MLLM, to generate several auxiliary attributes for each composition. Specifically, the following prompt is input to LLM: '*Please give me t adjectives that can describe the visual feature of a photo of a/an ... well.*', where $t$ is the number of auxiliary attributes and attribute-object composition (*e.g.*, peeled apple) is filled in '...'. Please refer to Appendix A for more details about the generation of auxiliary attributes by GPT-3.5. Subsequently, the generated auxiliary attribute words form a set $\mathbb{A}_a$. Therefore, the set of all words $\mathbb{Y}$ is obtained, including attributes, objects and auxiliary attributes as follows:

$$\mathbb{Y} = \mathbb{A} \cup \mathbb{O} \cup \mathbb{A}_a \tag{8}$$

**Obtaining MLLM embeddings for words and compositions.** Each word $y \in \mathbb{Y}$ is fed into LLaVA v1.5 to get the last hidden states, $i.e.$, $LLaVA_{lhs}(\cdot)$. Please refer to Appendix B for more details about the obtainment of the last hidden states of LLaVA v1.5 for an input word. Subsequently, they are passed through an MLP layer to get embeddings $E_{word}(\cdot)$ of aligned dimension with visual features. And for a composed pair $c$ of attribute $a$ and object $o$, $i.e.$, $c = (a, o)$, we get the last hidden states of LLaVA v1.5 for $a$ and $o$, respectively, which are then horizontally concatenated and fed into a linear layer $Lin_{comp}(\cdot)$ to get the composed pair embedding $E_{comp}(\cdot)$. The process is formulated as follows:

$$E_{word}(y) = MLP_{wd}(LLaVA_{lhs}(y)) \tag{9}$$

$$E_{comp}(c) = Lin_{comp}(Cat((LLaVA_{lhs}(a), (LLaVA_{lhs}(o)))) \tag{10}$$

**Word expanding.** Prior works compute cosine similarities of disentangled features and word embeddings only within the respective domains of attributes or objects, which results in the disentangled attributes and objects still retaining the information of each other. To address the problem, we propose a word expanding strategy, which computes cosine similarities of visual features and the embeddings of all words, including attributes and objects, and treats all words except the ground-truth word as negative labels.

**Alignment by cross-entropy.** Similar to Mancini et al. (2021), we use cross-entropy to process the cosine similarity of visual features and word embeddings. Assume that $\boldsymbol{f}$ is the visual embedding

and $E_{word}(wd)$ is the word embedding for the word $wd \in \mathbb{Y}$ in a joint space. The classifier logit from $\boldsymbol{f}$ to $E_{word}(wd)$ is defined as follows:

$$CE(\boldsymbol{f}, wd) = \frac{e^{\delta \cdot cos(\boldsymbol{f}, E_{word}(wd))}}{\sum_{y \in \mathbb{Y}} e^{\delta \cdot cos(\boldsymbol{f}, E_{word}(y))}} \tag{11}$$

where $\delta$ is the temperature variable, and $cos(\cdot, \cdot)$ denotes cosine similarity function. Thus cross-entropy with/without label smoothing can be uniformly formulated as follows:

$$H(\boldsymbol{f}, \mathbb{Y}) = \sum_{y \in \mathbb{Y}} -z \log(CE(\boldsymbol{f}, y)) \quad \text{with} \quad z = \begin{cases} 1 - \alpha, & \text{if } y \text{ is ground truth label} \\ \alpha/t, & \text{if } y \text{ is auxiliary label} \\ 0, & \text{otherwise} \end{cases} \tag{12}$$

where $\alpha$ denotes the smoothing factor, $t$ refers to the number of auxiliary labels and $z \in [0, 1]$ represents the target value of one-hot or smoothing label. For cross-entropy without label smoothing, *i.e.* with one-hot label $H_{oh}$, $\alpha$ and $t$ are set to 0. And the cross-entropy with label smoothing is denoted as $H_{ls}$.

For the disentangled attribute features of one image relative to each other, since a single object exhibits multiple attributes, we exploit attribute smoothing with auxiliary attributes to undermine the confidence in the ground-truth attribute and learn more related attributes. For the shared attribute features without relativity, one-hot label is used to compute its classification loss. The loss for disentangled attributes can be defined as follows:

$$\mathcal{L}_{attr} = \sum_{e \in \{m2a, a2m, m2o, o2m\}} H_{ls}(F_{attr}^{e}, \mathbb{Y}) + H_{oh}(F_{attr}^{ma}, \mathbb{Y}) \tag{13}$$

Concerning the disentangled object features, we use cross-entropy with one-hot label to learn the prototype of the object and the loss is as follows:

$$\mathcal{L}_{obj} = \sum_{e \in \{m2a, a2m, m2o, o2m\}} H_{oh}(F_{obj}^{e}, \mathbb{Y}) + H_{oh}(F_{obj}^{mo}, \mathbb{Y}) \tag{14}$$

With respect to the visual feature of the image from image embedder, we calculate the cosine similarity between visual embedding and the composed pair embedding of the corresponding composition label and use one-hot label to align them. The classification loss for compositions is as follows:

$$\mathcal{L}_{comp} = \sum_{img \in \{m, a, o\}} H_{oh}(F_{comp}^{img}, \mathbb{C}_s) \tag{15}$$

### 3.3 TRAINING AND INFERENCE

During the training phase, the overall loss function is formulated as follows:

$$\mathcal{L} = \gamma_{ortho} \mathcal{L}_{ortho} + \gamma_{comp} \mathcal{L}_{comp} + \gamma_{attr} \mathcal{L}_{attr} + \gamma_{obj} \mathcal{L}_{obj} \tag{16}$$

where $\gamma_{ortho}$, $\gamma_{comp}$, $\gamma_{attr}$, and $\gamma_{obj}$ are weighting factors to balance the influence of different losses.

For inference, we use the composition feature space generated by the classifier that is obtained by optimizing $\mathcal{L}_{comp}$. Specifically, given an image from test set, the cosine similarities of its visual feature obtained by image embedder and the composed pair embeddings of all candidate compositions in the test set are computed. The composition with the highest similarity is the class predicted by the model. Note that although the disentanglement branches are not used for inference, they still influence the formation of the composition feature space.

## 4 EXPERIMENT

### 4.1 EXPERIMENT SETUP

**Datasets.** We evaluate our model on three challenging CZSL benchmark datasets: MIT-states (Isola et al., 2015), C-GQA (Naeem et al., 2021), and VAW-CZSL (Saini et al., 2022). MIT-states consists

| Method | MIT-States | | | | C-GQA | | | | VAW-CZSL | | | |
|---|---|---|---|---|---|---|---|---|---|---|---|---|
| | *AUC* | *HM* | *Seen* | *Unseen* | *AUC* | *HM* | *Seen* | *Unseen* | *AUC* | *HM* | *Seen* | *Unseen* |
| SymNet (Li et al., 2020) | 3.2 | 13.7 | 22.7 | 20.1 | 1.9 | 10.8 | 20.3 | 11.8 | 2.8 | 13.5 | 20.2 | 18.0 |
| CompCos (Mancini et al., 2021) | 12.3 | 28.2 | 39.0 | 39.5 | 5.0 | 17.7 | 32.8 | 19.1 | 6.5 | 20.8 | 30.5 | 27.4 |
| Co-CGE (Mancini et al., 2022) | 10.3 | 25.1 | 41.0 | 33.1 | 4.2 | 15.2 | 32.9 | 17.0 | 6.2 | 19.7 | 31.0 | 26.1 |
| SCEN (Li et al., 2022) | 9.8 | 24.6 | 35.1 | 36.5 | 3.8 | 15.3 | 31.5 | 15.7 | 5.7 | 19.2 | 29.9 | 24.5 |
| OADis (Saini et al., 2022) | 13.1 | 29.0 | 42.3 | 27.3 | 2.3 | 12.1 | 23.3 | 12.8 | 4.1 | 16.2 | 26.0 | 20.7 |
| INV (Zhang et al., 2022b) | 11.5 | 26.6 | 28.5 | 25.0 | 1.4 | 7.9 | 28.6 | 6.8 | 2.0 | 11.1 | 21.1 | 11.9 |
| CANet (Wang et al., 2023) | 13.6 | 29.8 | **46.4** | 39.9 | 5.7 | 18.9 | 34.8 | 20.5 | 6.7 | 21.0 | 31.2 | 27.4 |
| ProCC (Huo et al., 2024) | 9.5 | 28.1 | 43.1 | 39.1 | 3.5 | 15.1 | 32.4 | 15.8 | 3.6 | 18.9 | 26.9 | 25.5 |
| CLIP (Nayak et al., 2023) | 11.0 | 26.1 | 30.2 | 46.0 | 1.4 | 8.6 | 7.5 | 25.0 | - | - | - | - |
| CoOp (Nayak et al., 2023) | 13.5 | 29.8 | 34.4 | **47.6** | 4.4 | 17.1 | 20.5 | **26.8** | - | - | - | - |
| **TRIDENT** (Ours) | **14.2** | **30.9** | 44.5 | 40.0 | **8.0** | **22.6** | **39.5** | 24.1 | **8.3** | **23.4** | **33.3** | **31.1** |

Table 1: Comparison with the state-of-the-art results on MIT-States, C-GQA and VAW-CZSL. We compare our **TRIDENT** with the state-of-the-art methods on test *AUC*, best seen (*Seen*), best unseen (*Unseen*) and best harmonic mean (*HM*) accuracies on these three datasets. We measure top-1 *AUC* on MIT-States and C-GQA, and top-3 *AUC* on VAW-CZSL. Best results are displayed in **boldface**, and second best results are underlined.

of diverse real-world images labeled automatically by early image search engine technology. C-GQA and VAW-CZSL are two more challenging benchmark datasets that consist of broad collections of in-the-wild images. C-GQA has more one-to-one attribute-object compositions, while objects in VAW-CZSL share more attributes. We present the common data splits of the three datasets in Appendix C.

**Metrics.** Following the common generalized CZSL setting (Purushwalkam et al., 2019), we evaluate the performance of our model on both seen and unseen pairs separately. Based on them, a calibration bias trades off between the prediction scores of seen and unseen pairs. We calculate area under the curve *AUC* (in %) using seen and unseen classification accuracies at different biases in test data. We also report the best seen and unseen accuracies *Seen* and *Unseen* (in %) of the curve. In addition, we calculate the harmonic mean of seen and unseen classification accuracies at difference bias and report the best one *HM* (in %).

**Implementation details.** We use the visual encoder of LLaVA v1.5, Vit-large-14-336px as our frozen feature extractor, whose outputs contain 577 tokens (1 [CLS] and 576 patch tokens) of 1024 dimensions. The cross-modal connector of LLaVA v1.5 maps the features to the dimension of 4096, the same as last hidden states of based LLM Vicuna v1.5 (Zheng et al., 2023). Image embedder and the MLP for words map them to the dimension of 1024 for faster training. **TRIDENT** and all baseline models are trained with 128 batch size for 50 epochs. The number of global features is set to 6, 2, 4 for the three datasets, respectively, and the number of local features is twice that of global features. The label smoothing factor is set to 0.09, 0.03, 0.03 for the three datasets, respectively. The number of generated auxiliary attributes for each composition is set to 3. Refer to Appendix D for more information about implementation.

**Baselines.** We compare our **TRIDENT** with recent and prominent approaches in the task of CZSL: SymNet (Li et al., 2020), CompCos (Mancini et al., 2021), Co-CGE (Mancini et al., 2022), SCEN (Li et al., 2022), OADis (Saini et al., 2022), INV (Zhang et al., 2022b), CANet (Wang et al., 2023), and ProCC (Huo et al., 2024). We replace their backbone with Vit-large-14-336px and retrain all models with the same epoch for the sake of fairness. In addition, we choose CLIP and CoOp as baselines for their strong zero-shot classification abilities.

## 4.2 RESULTS AND DISCUSSION

In this section, we compare **TRIDENT** with state-of-the-art methods. As shown in Table 1, **TRIDENT** surpasses other models by a substantial margin in general. For MIT-States, **TRIDENT** boosts *AUC*, *HM*, and *Unseen* from 13.6%, 29.8%, and 39.9% of CANet to the new state-of-the-art performance of 14.2%, 30.9%, and 40.0% with 0.6%, 1.1%, and 0.1% improvement, respectively. Due to the considerable label noise in the MIT-States benchmark (Atzmon et al., 2020), our model achieves comparable performance as compared to other baselines. However, for the more challenging benchmark C-GQA, **TRIDENT** achieves 8.0%, 22.6%, 39.5%, and 24.1% on the metrics of *AUC*, *HM*, *Seen*, and *Unseen*, providing 2.3%, 3.7%, 4.7%, and 3.6% improvements on the previous state-of-the-art model CANet. For the existing most challenging benchmark dataset VAW-

CZSL, **TRIDENT** attains performance of 8.3%, 23.4%, 23.4%, and 33.3%, surpassing CANet by 1.6%, 2.4%, 2.2%, and 3.7% in terms of *AUC*, *HM*, *Seen*, and *Unseen*. We observe that **TRIDENT** performs significantly better than CANet regarding all metrics on two challenging and low-noise benchmark dataset C-GQA and VAW-CZSL, which indicates the efficacy of our approach. This improvement arises from the utilization of MLLM embeddings and attribute smoothing, which enhance attribute-object disentanglement and consequently facilitate the recognition of unseen compositions while maintaining performance on seen compositions.

In addition, we compare **TRIDENT** with two pretrained Vision-Language Models (VLM), CLIP and CoOp, after fine-tuned for the CZSL task. Since they are trained on a large amount of image-text data, they possess zero-shot image classification capabilities, which leads to better classification results for unseen images. Regarding the metrics of *Unseen*, CoOp outperforms **TRIDENT** by 7.6% and 2.7% on MIT-States and C-GQA, respectively. However, **TRIDENT** surpasses CoOp by 0.7% and 1.1% on the core metrics of *AUC* and *HM* on MIT-States, as well as 3.6% and 5.5% on C-GQA, which suggests **TRIDENT** performs better than CLIP and CoOp in CZSL task.

| Method | *AUC* | *HM* | *Seen* | *Unseen* |
|---|---|---|---|---|
| *w/o condition_masks* | 13.4 | 30.1 | 44.3 | 39.7 |
| *w/o FAAs* | 12.9 | 28.9 | 42.6 | 38.0 |
| *w/o word_expanding* | 14.0 | 30.1 | 44.7 | 39.8 |
| *w/o attribute_smoothing* | 13.9 | 30.5 | **44.9** | 39.5 |
| *w/o $\mathcal{L}_{attr} + \mathcal{L}_{obj}$* | 13.2 | 30.1 | 43.8 | 38.9 |
| *w/o $\mathcal{L}_{ortho}$* | 14.1 | 30.7 | 44.6 | 39.7 |
| **TRIDENT** | **14.2** | **30.9** | 44.5 | **40.0** |

Table 2: Ablation study results on MIT-States. *w/o certain_part* denotes this part is ablated.

| Method | *Varient* | *AUC* | *HM* |
|---|---|---|---|
| SCEN | *ft+w2v* | 8.2 | 22.8 |
| | $LLaVA_{lhs}$ | **10.3** | **25.1** |
| CANet | *ft+w2v* | 12.3 | **28.4** |
| | $LLaVA_{lhs}$ | **12.5** | 28.3 |
| **TRIDENT** | *ft+w2v* | 14.0 | 29.9 |
| | $LLaVA_{lhs}$ | **14.2** | **30.9** |

Table 3: Impact of word embedding on MIT-States.

### 4.3 ABLATION STUDY

**Effectiveness of each component.** We ablate certain module of **TRIDENT** to evaluate the contribution of each module on MIT-States, as it is the most common used dataset. The ablation results are reported in Table 2. From this table, we gain the following observations.

1) Both *w/o condition_masks* model and *w/o FAAs* model perform worse than **TRIDENT**, which validates the importance of extracting the multi-granularity features and filtering out the background noise, respectively.

2) **TRIDENT** surpasses *w/o word_expanding* model and *w/o attribute_smoothing* model on the *Unseen* metric, yet falls short of them in the *Seen* metric. The difference between **TRIDENT** and the *w/o word_expanding* model on the two metrics stems from its more thorough disentanglement, which enhances the recognition of unseen images while weakens the identification of seen images. The disparity between **TRIDENT** and the *w/o attribute_smoothing* model arises from attribute smoothing, which diminishes the confidence of the model in seen compositions, facilitating its generalization to unseen compositions. However, the improvement of **TRIDENT** over these two models on *AUC* and *HM* indicates the effectiveness of word expanding and label smoothing strategy.

3) **TRIDENT** outperforms *w/o $\mathcal{L}_{attr} + \mathcal{L}_{obj}$* model on all metrics, confirming that the attribute-object disentanglement module is highly advantageous for generalization from seen compositions to unseen compositions.

4) w/o $\mathcal{L}_{ortho}$ model is inferior to **TRIDENT**, which suggests the designed orthogonal regularization is indeed helpful to guarantee that different features extract different information.

**Impact of word embeddings.** Our work leverages the last hidden states of LLaVA v1.5 ($LLaVA_{lhs}$) as word embeddings, while Word2Vec (Mikolov, 2013) and Fasttext (Bojanowski et al., 2017) are the most common word embeddings for MIT-States in previous works. In Table 3, based on three models: SCEN, CANet and **TRIDENT**, we compare the performance of employing the last hidden states of LLaVA v1.5 and the sum of Word2Vec and Fasttext (*ft+w2v*), respectively. The results in-

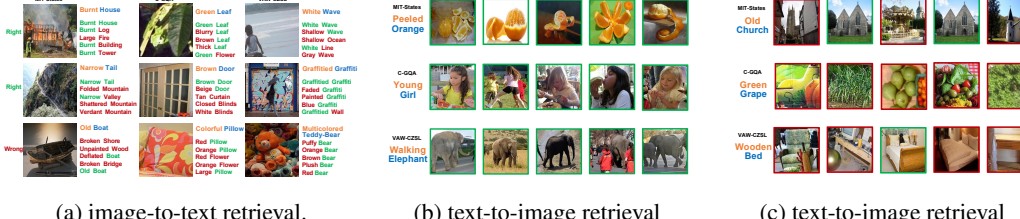

(a) image-to-text retrieval.

(b) text-to-image retrieval
(successful cases).

(c) text-to-image retrieval
(failure cases).

Figure 3: Qualitative analysis. (a) Top-5 image-to-text retrieval cases. The first two rows display successful cases, while the last row presents failure cases. (b) Successful cases of top-5 text-to-image retrieval. (c) Failure cases of top-5 text-to-image retrieval. In all cases, the attribute and object of composition label are marked in orange and blue, respectively. And the successful and failure retrieval results are tagged in green and red, respectively.

dicate that the last hidden states of MLLM capture more complex multimodal semantic information than ordinary word embeddings.

For details on the impact of hyperparameters, including the number of visual features and the label smoothing factor, please refer to Appendix E.

### 4.4 QUALITATIVE ANALYSIS

Inspired by Hao et al. (2023), we use **TRIDENT** to conduct both image-to-text retrieval and text-to-image retrieval experiments on the three datasets.

We first consider image-to-text retrieval, shown in Figure 3a. The first two rows display successful cases, while the last row presents failure cases. And the cases shown in these three columns are drawn from the three datasets, respectively. Given an image, such as the image of `burnt house`, we extract its visual features by image embedder and retrieve the top-5 closest composed pair embeddings of compositions. For successful cases, such as the image labeled `burnt house`, we notice that the top and other predictions can both describe the image. In this image, there are logs burning on fire, so the top-4 predictions of the image can also describe it. In terms of the image labeled `green leaf`, another successful case, the predicted attributes can also describe `leaf`, which is thanks to attribute smoothing learning more attributes for an object. For the failure cases, such as the image labeled `multicolored teddy-bear`, the model pays more attention to the main puffy and orange bear and neglects the background of multicolored bears.

We then consider text-to-image retrieval. Successful cases are shown in Figure 3b, while failure cases are shown in Figure 3c. Given a text composition, we embed it and retrieve the top-5 closest images. We can observe that the retrieved images of `peeled orange` are definitely correct. However, the retrieved images of `green grapes` are all wrong. This is due to the fact that the training images of `green grapes` in C-GQA dataset are almost filled with a single grape, making it difficult for the model to capture the contour features of a bunch of `green grapes`. The image-to-text and text-to-image retrieval experiments confirm that our model effectively projects visual features and word embeddings into a unified space.

## 5 CONCLUSION

In this work, we propose a novel framework termed MLLM embeddings and attribute smoothing guided disentanglement (**TRIDENT**) to address the challenging CZSL task. First, we leverage feature adaptive aggregation modules to mitigate the impact of background, and utilize learnable condition masks to capture multi-granularity features for attribute-object disentanglement. In addition, we exploit the last hidden states of MLLM to replace ordinary word embeddings, as they can capture complex multimodal semantic information. Moreover, we leverage LLM to generate auxiliary attributes and perform attribute smoothing to diminish overconfidence of models in seen compositions, which enables models to generalize to unseen compositions better. Extensive experiments have been conducted on three challenging datasets, and the results demonstrate the effectiveness of **TRIDENT**. In the future, we plan to extend our method to harness the powerful capabilities of LLMs, MLLMs, and CLIP to more effectively address the CZSL task.

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

# A   AUXILIARY ATTRIBUTES GENERATION BY LLM AND POST-PROCESS

As mentioned before, we leverage GPT-3.5 to generate some auxiliary attributes for attribute smoothing. The auxiliary attributes are generated based on the contextual composition of the object and its attribute, such as `ripe apple` or `sliced apple`. The model takes into account the unique characteristics that arise from the combination of attribute and the object's context. This ensures that the auxiliary attributes accurately capture the nuances of the specific composition, rather than general object-level attributes. Since the generation ability of LLM is affected by many factors, in this section, we first explore the impact of different prompts on LLM-generated content. Then we study on the influence of the number of auxiliary attributes $t$. In addition, we introduce post-processing of generated text to eliminate potential noise.

| | |
|---|---|
| User: | Please give me five adjectives that are most likely to describe the browned beef well. |
| GPT-3.5: | 1. Flavorful 2. Juicy 3. Savory 4. Tender 5. Rich |
| User: | Now 1 give you an object with an adjective: browned beef. Please give me five more possible adjectives that can describe the browned beef well. |
| GPT-3.5: | 1. Juicy beef 2. Tender beef 3. Flavorful beef 4. Savory beef 5. Succulent beef |
| User: | Please give me five common adjectives that can describe a browned beef well. |
| GPT-3.5: | 1. Juicy 2. Brown 3. Savory 4. Tender 5. Succulent |
| User: | Please give me five adjectives that can describe the visual feature of a photo of a/an broned beef well. |
| GPT-3.5: | 1. Juicy 2. Glistening 3. Crispy 4. Sizzling 5. Mouthwatering |

Table 4: Example prompts with the composition `browned beef` input to GPT-3.5 and the generated text.

| | |
|---|---|
| User: | Please give me five adjectives that are most likely to describe the inflated pool well. |
| GPT-3.5: | 1. Spacious 2. Durable 3. Sturdy 4. Bright 5. Relaxing |
| User: | Now 1 give you an object with an adjective: inflated pool. Please give me five more possible adjectives that can describe the inflated pool well. |
| GPT-3.5: | 1. Refreshing inflated pool 2. Relaxing inflated pool 3. Blue inflated pool 4. Large inflated pool 5. Fun inflated pool |
| User: | Please give me five common adjectives that can describe a inflated pool well. |
| GPT-3.5: | 1. Large 2. Round 3. Inviting 4. Fun 5. Sturdy |
| User: | Please give me five adjectives that can describe the visual feature of a photo of a/an inflated pool well. |
| GPT-3.5: | 1. Colorful 2. Refreshing 3. Sparkling 4. Bright 5. Relaxing |

Table 5: Example prompts with the composition `inflated pool` input to GPT-3.5 and the generated text.

**Impact of prompt input to LLM.** Users typically interact with LLMs through prompts to request answers to questions, generate text, complete tasks, and more. The model generates text based on the provided prompt, striving to meet the user's requirements (Sahoo et al., 2024). Therefore, the good design of prompt is significant for stimulating knowledge of LLMs, which enables them to better follow our instructions and generate auxiliary attributes with high quality. We first design some prompts with different style, then input them into GPT-3.5 and observe the quality of generated attributes. Some prompt examples on the composition `browned beef` and `ancient building` are shown in Table 4 and Table 5, respectively.

As shown in Table 4, the prompt without "the visual feature of ..." may cause the model to produce adjectives that are not specific but generic, such `Savory` and `Rich`. In both Table 4 and Table 5, the

prompts starting with "Now I give you...", compared to those starting with "Please give me ...", result in a weaker instruction following ability of the model. Therefore, we choose the prompt: "Please give me five adjectives that can describe the visual feature of a photo of a/an ... well."

**Impact of the number of auxiliary attributes** $t$**.** In Table 4, we observe that the generated attributes describe the compositions to varying degrees, with later items in the sequence being less relevant generally. Therefore, we study on the influence of the number of auxiliary attributes $t$.

| $t$ | the generated text for the composition `large garden` |
|---|---|
| 3 | 1. Lush 2. Vibrant 3. Flourishing |
| 5 | 1. Lush 2. Expansive 3. Vibrant 4. Serene 5. Verdant |
| 10 | 1. Lush 2. Vibrant 3. Expansive 4. Serene 5. Colorful 6. Beautiful 7. Bountiful 8. Captivating 9. Peaceful 10. Tranquil |

Table 6: Impact of $t$ on the generated text with the composition `large garden`. Note that the input prompt provided to GPT-3.5 is the previously selected one, replacing $t$ and the composition.

| $t$ | the generated text for the composition `young girl` |
|---|---|
| 3 | 1. Innocent 2. Radiant 3. Youthful |
| 5 | 1. Youthful 2. Innocent 3. Vibrant 4. Radiant 5. Captivating |
| 10 | 1. Radiant 2. Innocent 3. Vibrant 4. Captivating 5. Playful 6. Ethereal 7. Alluring 8. Charming 9. Enchanting 10. Happpy |

Table 7: Impact of $t$ on the generated text with the composition `young girl`. Note that the input prompt provided to GPT-3.5 is the previously selected one, replacing $t$ and the composition.

Table 6 and Table 7 show the generated text using different $t$ of compositions `large garden` and `young girl`. The results demonstrate that the greater the number, the more generic adjectives with irrelevant information are included, for example, `Captivating` is generated for both compositions. In addition, with $t$ increasing, the noise in the generated text due to the uncertainty of the model about the given image grows. The `young girl` may not be happy, yet the model fails to find ten words to describe her, so it has to guses. Therefore, we set $t$ to 3, this minimizes the general adjectives and noise while retaining useful information.

**Post-processing of generated text.** GPT-3.5 generates a segment of text, which we need to process into multiple useful words by exploiting regular expressions. However, the auxiliary attributes generated by LLM may contain the attribute of the input composition, for example, generating `ancient` for `ancient building`. At this point, we reuse the model to generate $t + 1$ adjectives for this composition and select three adjectives that are not the attribute of the input composition.

## B    OBTAINMENT OF THE LAST HIDDEN STATES OF MLLM

We input the attribute (object) word into LLaVA v1.5, which first tokenizes the word into z tokens. These tokens pass through all attention blocks in the MLLM, ultimately generating z embeddings of dimension $d_m$ after the last block, named the last hidden states. Subsequently, we apply average pooling to these z embeddings of dimension $d_m$ to obtain a $d_m$-dimensional embedding that represents the attribute. Since the last hidden states are designed to generate the next token rather than for representation, Muennighoff et al. (2024) leverages instruction to fine-tune the model. Therefore, we fine-tune the last hidden states with a low learning rate during the training phase of **TRIDENT**.

## C   DATA STATICTICS

Table 8 shows detailed data statistics following the common data splits of MIT-States (Isola et al., 2015), C-GQA (Naeem et al., 2021) and VAW-CZSL (Saini et al., 2022). MIT-States contains 53753 images, with 115 attributes and 245 objects. It comprises 1262 seen compositions and 300/400 (validation/test) unseen compositions. C-GQA is a natural image dataset which contains 39298 images, with 413 attributes and 764 objects. It includes 5,592 seen compositions and 1,040/923 (validation/test) unseen compositions. VAW-CZSL is a larger dataset which contains 440 attributes and 541 objects for 238040 images, and it is split into 11175 seen and 2322/2470 unseen compositions for training and validation/testing, respectively.

| | Composition | | | Train | | Validation | | | Test | | |
| Dataset | $|\mathbb{A}|$ | $|\mathbb{O}|$ | $|\mathbb{A} \times \mathbb{O}|$ | $|\mathbb{C}_s|$ | $|\mathbb{X}|$ | $|\mathbb{C}_s|$ | $|\mathbb{C}_u|$ | $|\mathbb{X}|$ | $|\mathbb{C}_s|$ | $|\mathbb{C}_u|$ | $|\mathbb{X}|$ |
|---|---|---|---|---|---|---|---|---|---|---|---|
| MIT-States (Isola et al., 2015) | 115 | 245 | 28175 | 1262 | 30338 | 300 | 300 | 10420 | 400 | 400 | 12995 |
| C-GQA (Naeem et al., 2021) | 413 | 674 | 278362 | 5592 | 26920 | 1252 | 1040 | 7280 | 888 | 923 | 5098 |
| VAW-CZSL (Saini et al., 2022) | 440 | 541 | 238040 | 11175 | 72203 | 2121 | 2322 | 9524 | 2449 | 2470 | 10856 |

Table 8: Summary statistics of the datasets used in our experiments.

## D   IMPLEMENTATION DETAILS

We use NVIDIA PTX 3090 GPU to train all models under the Pytorch framework (Paszke et al., 2019). Since $\mathcal{L}_{comp}$ leverages image features during training, we use a Batch Normalization, ReLU and 0.3 dropout for Image embedder. We train **TRIDENT** by Adam optimizer with weight decay 5e-5, learning rates 1.5e-6 for word embedding as well as 2e-4 for other modules on three datasets. We decay the learning rate by 10 at epoch 30 and 40. The temperature variable of cosine similarity $\delta$ is set to 0.05. For weighting coefficients $\gamma_{ortho}, \gamma_{comp}, \gamma_{attr}, and \gamma_{obj}$, we set them to 0.1, 1, 0.5, and 0.5, respectively.

## E   IMPACT OF HYPERPARAMETERS

To provide more insight into the effect of visual features and label smoothing, we study on the performance of TRIDENT with respect to different numbers of visual features and different label smoothing factors, respectively. Experiments exploring the impact of hyperparameters are conducted on datasets MIT-States and C-GQA.

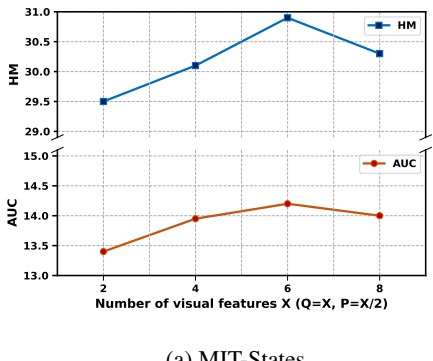
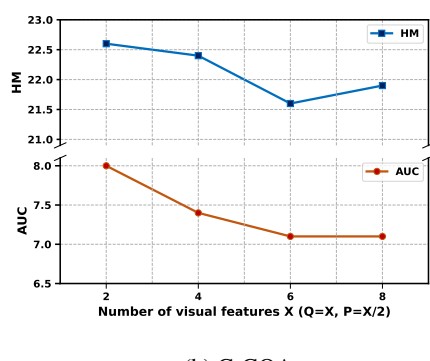

(a) MIT-States                                        (b) C-GQA

Figure 4: Impact of the number of the visual features on (a) MIT-States and (b) C-GQA.

**Impact of the number of visual features.** We illustrate the performance of TRIDENT influenced by different numbers of attribute features in Figure . In Figure 4a, the performance of our model on MIT-States generally improves with the increasing number of visual features, but subsequently declines. This trend is reasonable, as a greater number of Visual features contains more useful information, thereby enhancing the performance. However, the number of useful features is limited; thus,

an excessive number of visual features may introduce redundancy and noise, ultimately hampering the performance of the model.

However, in Figure 4b, as the number of visual features increases, the performance of the model on C-GQA tends to decline overall. This may be attributed to the model's strong expressive capability in handling composition reasoning. In the low-noise C-GQA dataset, optimal performance can be achieved using only two features. Increasing the number of features, however, results in heightened model complexity without tangible benefits, potentially impairing generalization to unseen compositions. In contrast, the MIT-States dataset exhibits significant noise; thus, while the increase of visual features may introduce more noise, it also necessitates a greater amount of useful information, which can effectively mitigate the impact of the noise.

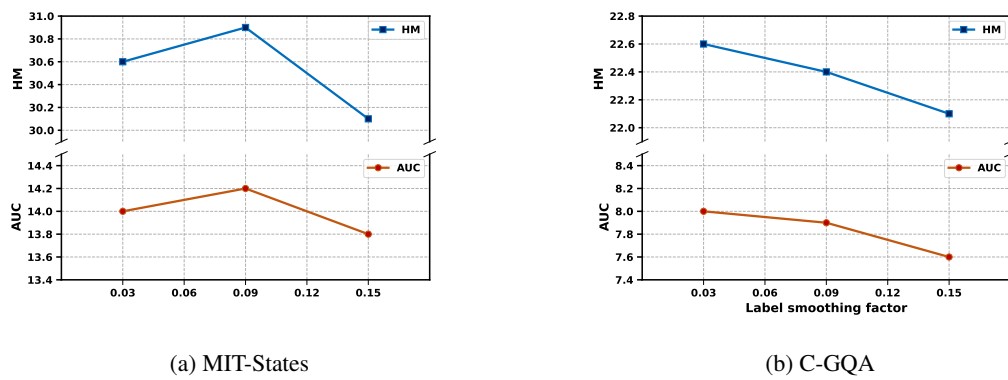

(a) MIT-States                                          (b) C-GQA

Figure 5: Impact of the label smoothing factor on (a) MIT-States and (b) C-GQA.

**Impact of the number of label smoothing factor.** The label smoothing factor $\alpha$ modulates the extent to which the model's confidence in seen compositions is attenuated. Figure 5a shows that as $\alpha$ increases, the model's performance on MIT-States initially improves before subsequently declining. This is because if alpha is too small, label smoothing fails to enhance generalization, while if alpha is too large, it adversely affects the model's ability to learn the representation of the original labels, resulting in more losses than gains. However, as shown in Figure 5b, the model achieves the best performance with C-GQA a smaller $\alpha$. This may be attributed to the fact that, compared to everyday objects, LLMs are less familiar with in-the-wild objects, leading to relatively lower quality in the generated auxiliary attributes; thus, a smaller smoothing factor can mitigate the impact.

