# OpenReview forum: "Leveraging MLLM Embeddings and Attribute Smoothing for Compositional Zero-Shot Learning"
_ICLR.cc/2025/Conference — ICLR 2025 Conference Withdrawn Submission_

### Official Review · Reviewer_B7DA · 2024-10-27

**Soundness:** 3
**Presentation:** 2
**Contribution:** 2
**Rating:** 5
**Confidence:** 4

**Summary:**

In this paper, the authors propose a novel approach that leverages multimodal large language models (MLLM) and large language models (LLM) to predict the state-object pair for compositional zero-shot learning (CZSL). Moreover, attribute-object disentanglement and feature alignment are used to improve the primitive feature representations.

**Strengths:**

**1.** This work innovatively leverages multimodal large language model for CZSL, the idea is novel.

**2.** The organization of this article is reasonable and well-written.

**3.** Extensive experiments on three benchmarks show that the improvement in performance is noteworthy.

**Weaknesses:**

**1.** The Figure 2 is ambiguous: the training and frozen modules are not clearly labeled, for example, the last hidden states of MLLM is trained but not the LLM, and the image embedder is trained but not the visual backbone; the graphical representation is inconsistent, for example, the network module image embedder is represented by a rectangle, but FAA and MLP are represented by text lines, which can easily be confused with other text such as “patches”; in the attribute-object disentanglement stage, some MLPs are not labeled.

**2.** Some expressions are not accurate enough. For example, equation (12) uniformly represents with/without label smoothing, but $a/t$ is incorrect when $a$ and $t$ are both 0.

**3.** The method section of this paper devoted a great deal of space to introducing attribute-object disentanglement and feature alignment, but these modules are not used in the final inference process. So, how can the author's proposed modules be helpful for the final prediction?

**4.** Why isn't there a performance comparison with the latest works in 2024, such as Troika?

**Questions:**

Pleas see "Weaknesses"

---

### Official Review · Reviewer_Eecx · 2024-11-02

**Soundness:** 2
**Presentation:** 3
**Contribution:** 2
**Rating:** 5
**Confidence:** 4

**Summary:**

The paper introduces TRIDENT, a framework designed to improve Compositional Zero-Shot Learning (CZSL) by effectively disentangling attributes and objects in image compositions. By leveraging Multimodal Large Language Model (MLLM) embeddings and a unique attribute smoothing approach, TRIDENT addresses limitations in previous models, such as poor handling of background influence, lack of complex multimodal information in word embeddings, and overconfidence in seen compositions. TRIDENT employs adaptive feature aggregation modules, learns multi-granular features, and aligns visual features with embeddings from the last hidden states of MLLMs. Experiments demonstrate state-of-the-art performance on challenging datasets like MIT-States, C-GQA, and VAW-CZSL.

**Strengths:**

1. This paper introduces a novel method of attribute-object disentanglement with adaptive aggregation and learnable masks.
2. The framework’s effectiveness is substantiated through extensive experiments.
3. Attribute smoothing using auxiliary attributes generated by LLM shows promise in reducing overconfidence and enhancing model generalization.

**Weaknesses:**

1. Leveraging MLLMs like LLaVA to extract attribute embeddings raises potential concerns of data leakage, especially if the LLaVA model was trained on images from unseen pairs. This could inadvertently influence performance in the zero-shot setting.

2. While the paper claims to address overconfidence in seen compositions, Table 1 suggests that the primary performance improvements are concentrated in the seen classes, which appears to contradict this claim.

3. The performance gains over previous state-of-the-art models are modest. For example, on the MIT-States dataset, the HM metric only improves by 1.1% over the CoOp model. Additionally, it would be beneficial for the authors to report results on the UT-Zappos dataset, as it is commonly included in other works.

4. Finally, the ablation studies in Table 2 indicate that the model components individually contribute only marginal gains, suggesting that the impact of each module might be limited.

**Questions:**

Please refer to the weaknesses section.

---

### Official Review · Reviewer_6sdT · 2024-11-03

**Soundness:** 2
**Presentation:** 2
**Contribution:** 2
**Rating:** 5
**Confidence:** 3

**Summary:**

This paper proposes a method named TRIDENT for compositional zero-shot learning (CZSL). The approach includes a visual feature extraction model designed to capture both global and local features. Additionally, an Attribute-Object Disentanglement module is introduced to learn separate, disentangled representations of attributes and objects. To address the issue of overconfidence in seen compositions, the paper further introduces a feature alignment module aimed at enhancing generalization.

**Strengths:**

1. The paper is well-organized and easy to follow.

2. This paper conducts comprehensive research on CZSL, with a clear and straightforward motivation.

**Weaknesses:**

1. Some annotations could be simplified. For example, in Eq. (5), certain parts of the equation appear to be duplicated. Simplifying these would improve clarity.

2. In Section 3.2.2, the approach of using a weighted disentanglement module to separate object and attribute features, while elegant, is somewhat difficult to follow. Adding a small figure to illustrate the mechanism would enhance understanding. Additionally, this section provides limited evidence to demonstrate that these designs are effective and genuinely learn disentangled features.

3. The method does not consistently achieve the best results across all datasets, which suggests that it may lack robustness.

**Questions:**

Please see the points under Weaknesses above.

---

### Official Review · Reviewer_iwb1 · 2024-11-04

**Soundness:** 2
**Presentation:** 3
**Contribution:** 3
**Rating:** 5
**Confidence:** 4

**Summary:**

This paper proposes a novel framework named MLLM embeddings and attribute smoothing guided disentanglement framework (TRIDENT) for CZSL. The framework first uses feature adaptive aggregation modules to reduce the impact of image background noise, and then uses learnable condition masks to capture multi-granularity features for attribute-object disentanglement. In addition, the framework leverages the last hidden states of MLLM to replace the original word embeddings, as they capture more complex multimodal semantic information. Moreover, the framework uses a large language model to generate auxiliary attributes and reduces the model's overconfidence in seen compositions through attribute smoothing, making the model's generalization ability for unseen combinations better. This paper conducts extensive experiments on three datasets, and the experimental results demonstrate the effectiveness of the proposed framework.

**Strengths:**

1. This paper considers the impact of the background noise of the CZSL datasets, which is a critical problem, and proposes a solution using feature adaptive aggregation (FAA) modules. Ablation experiments show that the module is effective.
2. This paper points out the problem that objects in CZSL datasets naturally have multiple attributes, while there is only one label, which is also critical, and uses a large language model to solve this problem. Attribute smoothing is also proposed. By using the ability of a large language model, some auxiliary attributes associated with the current combination are generated. The one-hot label of the attribute is innovatively changed to attribute smoothing, which is reasonable for reducing the overfitting of model training.

**Weaknesses:**

1. Since the author believes that word embeddings, such as Word2Vec (Mikolov, 2013) and GloVe (Pennington et al., 2014) have a poor ability to capture cross-modal information, why not use CLIP  (Nayak et al., 2023)? CLIP is trained on image-text pairs and thus can solve this problem.
2. Missing comparisons with several recent papers [1,2,3] which are based on CLIP. CLIP-based methods outperform TRIDENT in Table 1. Comparative experiments between using the last hidden states of MLLM as word embedding and using CLIP should be added.
3. Introducing LLMs and MLLMs makes the comparisons between the proposed method and other methods somewhat unfair.

[1] Zheng, Zhaoheng, Haidong Zhu, and Ram Nevatia. "CAILA: Concept-Aware Intra-Layer Adapters for Compositional Zero-Shot Learning." Proceedings of the IEEE/CVF Winter Conference on Applications of Computer Vision. 2024.
[2] Jing, Chenchen, et al. "Retrieval-Augmented Primitive Representations for Compositional Zero-Shot Learning." Proceedings of the AAAI Conference on Artificial Intelligence. Vol. 38. No. 3. 2024.
[3] Bao, Wentao, et al. "Prompting language-informed distribution for compositional zero-shot learning." Proceedings of the European Conference on Computer Vision.

Minors:
1. In Figure 2, “aobj” should be “obj”.
2. In Eq.(13), does the first H_{oh} refer to H_{ls}?
3. The text in Figure 3 (a) is too small.

**Questions:**

See the weaknesses.

---

### Official Review · Reviewer_RyTd · 2024-11-06

**Soundness:** 3
**Presentation:** 3
**Contribution:** 2
**Rating:** 3
**Confidence:** 4

**Summary:**

This paper presented a method for Compositional Zero-Shot Learning. The main components of the method include: 1)  a feature adaptive aggregation modules to reduce the impact of background 2) an attribute-object disentanglement module by using both LLM and MLLM. 3) A label smoothing module to reduce the impact of excessive confidence in seen compositions. Experiments show some good results.

**Strengths:**

1) A simple pipeline that works well for the problem of Compositional Zero-Shot Learning.
2) Experiment results are good compared with some existing methods.

**Weaknesses:**

1) Novelty is very limited. The pipeline consists of three modules: feature extractor/aggregator; the so-called Attribute-Object Disentanglement by using a LLM to generate some potential adjective attributes; and feature alignment.
The only novelty is the use of LLM to generate potential attributes. This is to me somewhat very simple. While I understand it might lead to better generaization by using LLM than to train an attribute classifier as seen in the literature, this is very simple.
2) It's unclear to me which component of the pipeline contributes most to the final performance. More ablation experiments are needed

**Questions:**

see weakness

---

### Note · Authors · 2024-11-13

I have read and agree with the venue's withdrawal policy on behalf of myself and my co-authors.